# The Multifaceted Role of Growth Differentiation Factor 15 (GDF15): A Narrative Review from Cancer Cachexia to Target Therapy

**DOI:** 10.3390/biomedicines13081931

**Published:** 2025-08-08

**Authors:** Daria Maria Filippini, Donatella Romaniello, Francesca Carosi, Laura Fabbri, Andrea Carlini, Raffaele Giusti, Massimo Di Maio, Salvatore Alfieri, Mattia Lauriola, Maria Abbondanza Pantaleo, Lorena Arribas, Marc Oliva, Paolo Bossi, Laura Deborah Locati

**Affiliations:** 1Medical Oncology, IRCCS Azienda Ospedaliero-Universitaria di Bologna, 40138 Bologna, Italy; francesca.carosi3@studio.unibo.it (F.C.);; 2Department of Medical and Surgical Sciences (DIMEC), Alma Mater Studiorum, Università di Bologna, 40138 Bologna, Italy; 3Medical Oncology Unit, Sant’Andrea Hospital of Rome, 00189 Rome, Italy; 4Department of Oncology, University of Turin, Azienda Ospedaliero-Universitaria Città della Salute e della Scienza di Torino, 10126 Turin, Italy; 5Head and Neck Medical Oncology Department, Fondazione IRCCS Istituto Nazionale dei Tumori di Milano, 20133 Milan, Italy; 6Clinical Nutrition Unit, Catalan Institute of Oncology (ICO), L’Hospitalet de Llobregat, 08908 Barcelona, Spain; 7Bellvitge Biomedical Research Institute (IDIBELL), L’Hospitalet de Llobregat, 08908 Barcelona, Spain; 8Department of Medical Oncology, Catalan Institute of Oncology (ICO), L’Hospitalet de Llobregat, 08001 Barcelona, Spain; 9Department of Medical Oncology and Hematology, IRCCS Humanitas Research Hospital, Via Manzoni 56, 20089 Rozzano, Italy; 10Department of Internal Medicine and Medical Therapy, University of Pavia, 27100 Pavia, Italy; 11Medical Oncology Unit, Istituti Clinici Scientifici Maugeri IRCCS, 27040 Pavia, Italy

**Keywords:** GDF15, cancer biomarker, GFRAL, chemotherapy resistance, head and neck cancer, cachexia

## Abstract

**Background: **Growth Differentiation Factor 15 (GDF15) has emerged as a key biomarker and therapeutic target in oncology, with roles extending beyond cancer cachexia. Elevated GDF15 levels correlate with poor prognosis across several solid tumors, including colorectal, gastric, pancreatic, breast, lung, prostate, and head and neck cancers. GDF15 modulates tumor progression through PI3K/AKT, MAPK/ERK, and SMAD2/3 signaling, thereby promoting epithelial-to-mesenchymal transition, metastasis, immune evasion, and chemoresistance via Nrf2 stabilization and oxidative stress regulation. **Methods:** We performed a narrative review of the literature focusing on the role of GDF15 in solid tumors, with a particular emphasis on head and neck cancers. **Results:** In head and neck squamous cell carcinoma (HNSCC), GDF15 overexpression is linked to aggressive phenotypes, radioresistance, poor response to induction chemotherapy, and failure of immune checkpoint inhibitors (ICIs). Similar associations are observed in colorectal, pancreatic, and prostate cancer, where GDF15 contributes to metastasis and therapy resistance. Targeting the GDF15-GFRAL axis appears therapeutically promising: the monoclonal antibody ponsegromab improved cachexia-related outcomes in the PROACC-1 trial, while visugromab combined with nivolumab enhanced immune response in ICI-refractory tumors. **Conclusions:** Further investigation is warranted to delineate the role of GDF15 across malignancies, refine patient selection, and evaluate combinatorial approaches with existing treatments.

## 1. Introduction

Cancer cachexia (CC) is a multifactorial syndrome characterized by a progressive loss of skeletal muscle mass, with or without accompanying fat loss, that cannot be fully reversed by conventional nutritional support and leads to functional decline. The most widely accepted definition, proposed by Fearon et al. [1], includes weight loss greater than 5% over six months, or greater than 2% in individuals with low body mass index (BMI) (<20 kg/m^2^) or sarcopenia.

More recently, CC has been defined as unintentional depletion of muscle and adipose tissue, systemic inflammation, reduced appetite, and impaired energy metabolism [2]. CC affects up to 80% of patients with advanced malignancies, especially those with metastatic disease, and contributes to approximately 25% of cancer-related deaths [3,4]. Malnutrition and cachexia are associated with poorer survival and reduced tolerance to chemotherapy, increasing the risk of severe treatment-related toxicities [5]. A recent meta-analysis also linked CC with inferior outcomes and shorter time to treatment failure in patients receiving immune checkpoint inhibitors (ICIs), suggesting a detrimental impact on immunotherapy efficacy [6]. Nutritional impairment further correlates with reduced quality of life in head and neck, gastrointestinal, lung, and gynecological cancers [7] and may affect the tolerability of analgesic treatments such as transdermal fentanyl [8].

Given its clinical relevance, the European Society for Clinical Nutrition and Metabolism (ESPEN), in accordance with the Global Leadership Initiative on Malnutrition (GLIM), proposed consensus criteria requiring at least one phenotypic indicator (unintentional weight loss > 5% in 6 months, BMI < 20 kg/m^2^, or reduced muscle mass) and an etiologic criterion such as chronic systemic inflammation [3,9]. Basic assessment includes evaluation of muscle mass, inflammatory status (e.g., Glasgow prognostic score (GPS) based on C-Reactive Protein (CRP) and serum albumin), appetite, and muscle strength [4]. CC progresses through pre-cachexia, cachexia, and refractory cachexia [1]. Pre-cachexia involves <5% weight loss and is characterized by systemic inflammation, oxidative stress, altered glycogenolysis and gluconeogenesis, and early reductions in myogenesis and muscle contractility [4,10]. Established cachexia features a shift toward protein catabolism: amino acids are mobilized for hepatic gluconeogenesis, while insulin resistance promotes proteolysis and lipolysis via inhibition of the phosphoinositide 3-kinase (PI3K)/protein kinase B (PKB/AKT) pathway and activation of proteasomal degradation [11,12,13]. Refractory cachexia exacerbates these abnormalities and is associated with hypoalbuminemia, elevated CRP levels, and severe systemic complications, including cardiac atrophy, arrhythmias, thromboembolic events, respiratory problems, dysphagia, gastrointestinal atrophy, poor wound healing, and sepsis [14,15].

Current research focuses on identifying biomarkers in CC to improve diagnosis, prognostication, and the development of target therapies. Biomarkers include inflammatory cytokines, markers of muscle degradation, metabolic indicators, and regulators of neuroendocrine function [16]. Elevated levels of pro-inflammatory cytokines, such as tumor necrosis factor-α (TNF-α), interleukin-6 (IL-6), CRP, and monocyte chemoattractant protein-1 (MCP-1), are linked to systemic inflammation, neuroinflammatory changes, and muscle wasting, although their clinical utility is limited by inter-patient variability [17,18,19]. Transforming growth factor (TGF-β) family members, including activin A, myostatin, and growth differentiation factor 11 (GDF-11), inhibit muscle growth via activin type IIB receptor signaling [20,21], with activin A showing promise as a diagnostic [22] and prognostic marker of poor survival [23,24]. Markers of protein catabolism, such as 3-methylhistidine and fragments of titin and collagen, reflect negative protein balance but are influenced by diet and technical variability [25,26,27]. Metabolic dysregulation, including insulin resistance and altered lipid metabolism, contributes to reduced energy efficiency and progressive wasting [12,13]. Emerging microRNAs (e.g., miR-21, miR-486) may serve as non-invasive biomarkers, though validation is still needed and not yet feasible in daily clinical practice [28,29,30]. Adipose tissue-derived markers, including leptin, free fatty acids, and glycerol, indicate enhanced lipolysis but have limited diagnostic value due to confounding factors [31,32]. Zinc-α2-glycoprotein (ZAG), a lipolytic factor upregulated in CC, correlates with adipose depletion and weight loss, though further studies are required to establish its reliability relative to established mediators such as IL-6 and TNF-alpha [33,34].

Management strategies include nutritional support, pharmacologic treatments, exercise programs, and emerging targeted therapies. While nutritional supplementation and agents like megestrol acetate can stimulate appetite and weight gain, their effects are usually modest and short-lived, with little impact on survival. Corticosteroids may offer temporary symptom relief but are limited by significant adverse effects over time. Multimodal approaches combining diet, exercise, and medications are recommended but often face adherence challenges in advanced disease. More recently, targeted interventions, such as activin type II receptor inhibitors, have shown encouraging results in early studies but remain investigational [35].

Despite growing research efforts, no single biomarker has proven sufficient for diagnosis, prognosis determination, or therapeutic targeting, highlighting the complex, multifactorial nature of CC. In this context, growth differentiation factor-15 (GDF-15) has emerged as another potentially promising biomarker and interestingly as a potential therapeutic target addressing the underlying mechanisms of CC.

In this narrative review, we aim to synthesize the current evidence on the biological function of GDF-15 in physiological and pathological states, and to examine its role as a diagnostic, prognostic, and predictive biomarker across cancer types. Additionally, we discuss the therapeutic potential of GDF-15 inhibitors, either as monotherapy or in combination with other treatments, with a particular focus on head and neck cancers (HNC) as an example of malignancies with a high risk of malnutrition and cachexia due to both disease burden and treatment-related factors.

## 2. Physiological and Pathological Functions of GDF15

GDF15 was discovered in 1997 when Bootcov and colleagues cloned this molecule from a cellular line of human monocytoid cells and from then on, the potential roles and functions of GDF15 have gained increasing interest in the scientific community for its polyhedral implications in both pathological and physiological conditions [36,37]. Figure 1 illustrates the various roles of GDF15. The broad spectrum of functions of GDF15 is reflected by the evolution of its nomenclature over the years. The immunoregulatory effect mediated by the lipopolysaccharide-induced macrophage activation via the inhibition of TNF-alpha production was the first biological activity attributed to GDF-15. For this reason, it has also been referred to as macrophage inhibitory cytokine-1 (MIC-1) [36]. Since then, GDF15 has been identified by various names, including nonsteroidal anti-inflammatory drug-activated gene-1 (NAG-1), placental transforming growth factor beta (PTGFB), prostate-derived factor (PDF), and placental bone morphogenetic protein (PLAB) [38,39,40,41].

GDF15 is a member of the transforming growth factor β (TGF-β) superfamily, which encompasses several structurally related proteins involved in cellular growth and proliferation, differentiation, intercellular interactions and adhesion, migration, and cell death [48,49,50,51]. Nevertheless, GDF15 shares only a few features with TGF-β superfamily proteins, and it could be considered a separate, single-member family; it has retained the three-dimensional structure with the presence of a conserved cysteine sequence, which represents an essential component for protein stabilization [52,53,54]. GDF15 is encoded by a two-exon gene located on chromosome 19 at the 19p13.1 locus [55]. It is first secreted as a functionally inactive propeptide, known as pro-GDF15, which is composed of an N-terminal propeptide, which is a signal peptide that mediates trafficking and storage, and a C-terminal mature form [53,56]. The stepwise process of proteolytic cleavage generates the mature form of GDF-15 that circulates as a dimer. Therefore, GDF15 exists in three different forms: a pro-GDF15 monomer, a pro-GDF15 dimer, and a mature GDF15 homodimer. While the intermediate precursors are anchored to the extracellular matrix (ECM), the mature form has reduced affinity for ECM and can be detected in the circulation [57,58]. The only binding partner for GDF15 found to date is the glial cell-derived neurotrophic factor (GDNF) family receptor alpha-like protein (GFRAL) [59,60,61]. The GDF15/GFRAL pathway has increasingly emerged as an important regulator of energetic homeostasis, obesity, and cachexia (Figure 1). GFRAL is a transmembrane protein with a short cytoplasmic domain that belongs to the GDNF receptor α (GRFα) family, and it is encoded by a 9-exon gene located on chromosome 6p12. Currently, it represents a unique receptor with proven affinity for GDF15 [59,60,61,62,63]. Unlike GDF15, which is expressed in different tissues, GFRAL has a peculiar topography with a high concentration in the central nervous system (CNS) and, specifically, in correspondence to the area postrema (AP) and the nucleus of the solitary tract (NTS), cerebral areas involved in appetite regulation [59,60,61,62]. Lower concentrations of GFRAL have also been detected in human testes and adipose tissue. Recent evidence has also demonstrated the expression of GFRAL in prostate cancer cells, suggesting a potential role of GDF15 in disease pathogenesis [64]. The site-dependent distribution in the human body is probably the most responsible for the late recognition of this receptor. However, due to the variety of its functions, GDF15 is believed to interact with other still unknown receptors at distant sites and outside of the prostate and hindbrain [65]. The interaction between GDF15 and GFRAL activates several molecular pathways, as shown in Figure 2.

In physiological conditions, the serum concentration of GDF15 ranges from 200 to 1200 pg/mL in humans, but it can be subject to fluctuations in response to various stimuli [66]. It is found elevated in both physiological (such as elderly age, male sex, intense physical exercise, and smoking), iatrogenic (chemotherapy, radiation therapy, metformin, and anti-inflammatory drugs), and pathological conditions (infectious status, metabolic disorders, mitochondrial impairment, starvation, diabetes and insulin resistance, cardiovascular and renal diseases, chronic obstructive pulmonary disease, and tissue damage) [67,68,69,70,71,72,73,74].

In the case of neoplasms, GDF15 concentrations can significantly rise reaching values ranging between 10.000 and 100.000 pg/mL [68,75]. Genetic variants and polymorphisms have been called into question in the baseline determination of GDF15 concentrations [76].

Preliminary data about the involvement of GDF15 in starvation modulation and body weight regulation dates back to 2007 when researchers proved that the injection of a GDF15-overexpressing prostate cancer cell line in xenograft models resulted in the onset of cancer cachexia syndrome, with progressive weight loss directly proportional to the blood concentration of GDF15 [77]. These results were confirmed in humans, highlighting a correlation between gradually reducing BMI and GDF15 levels [77]. In synthesis, GDF15 can be regarded as both a hormone and a stress-induced cytokine, and its pivotal role in metabolic regulation has earned GDF15 the appellation of “satiety factor” [78].

Thus, GDF15 represents a promising prognostic or predictive biomarker, and further prospective studies are required to validate its relevance related to different clinical conditions including pre-cachexia or cachexia.

## 3. The Role of GDF15 in Various Tumor Types

In recent years, GDF15 has attracted considerable attention due to its involvement in carcinogenesis and CC. Elevated serum and/or tissue GDF15 levels have been consistently observed in several malignancies [79,80,81,82,83,84,85]. However, its role in carcinogenesis remains controversial, as both pro-tumorigenic and anti-tumorigenic functions have been described across different malignancies (Table 1).

Increased serum levels of GDF15 have been significantly associated with a higher number of colorectal adenomas and an elevated risk of recurrence for both adenomas and colorectal cancer (CRC) [86,87]. In addition, Li et al. demonstrated in vitro and in vivo that GDF15 promotes epithelial-to-mesenchymal transition (EMT) and metastasis via autocrine and paracrine pathways by activating Smad2 and Smad3 through the TGF-β type I receptor [79]. In CRC, GDF15 expression correlates with the occurrence of liver metastasis after surgical treatment and with poor survival outcomes [88]. Furthermore, GDF15 regulates chemosensitivity to oxaliplatin in CRC cells by reducing oxidative stress, which oxaliplatin exploits to induce tumor cell apoptosis. GDF15 post-transcriptionally stabilizes nuclear factor erythroid 2-related factor 2 (Nrf2), a master regulator of reactive oxygen species (ROS) levels, via the PI3K/AKT/glycogen synthase kinase-3β (GSK-3β) pathway, creating a positive feedback loop that enhances redox homeostasis and chemoresistance [89]. However, Yamaguchi et al. reported that GDF15 exerts anti-tumorigenic and pro-apoptotic effects in CRC cell lines, with its expression regulated by the PI3K/AKT/GSK-3β pathway, where AKT inhibition of GSK-3β promotes GDF15 expression [90].

In gastric cancer (GC), serum GDF15 levels exhibit diagnostic and prognostic value. GDF15 promotes tumor cell proliferation and invasion by upregulating STAT3 phosphorylation [80] or by activating ErbB2, which is central to the PI3K/AKT and mitogen-activated protein kinase (MAPK) pathways [91]. A combination of biomarkers [GDF15, matrix metalloproteinase-7 (MMP7), and miR-200c] has been associated with adverse outcomes in GC patients [92]. Elevated serum GDF15 levels have been implicated in mitochondrial dysfunction and chemoresistance to cisplatin in GC, possibly through enhanced antioxidant activity [93,94]. Conflicting data have been reported regarding the association between serum and tissue levels of GDF15 and overall survival in GC [92,95]: high tissue levels have been detected in GC cells, correlating with differentiation grading. However, it has been suggested that GDF15 upregulation may exert a tumor-suppressive role during the early phases of tumorigenesis, while later acquiring the capacity to promote tumor progression. Therefore, its utility as a prognostic biomarker remains controversial.

In pancreatic cancer (PC), the autocrine GDF15-GFRAL axis drives tumor growth and metastasis [96], promoting progression and therapeutic resistance through pathways involving nuclear receptor NR5A2 [97] and p38 MAPK [98]. Additionally, biomechanical compression-induced secretion of GDF15 by pancreatic fibroblasts fosters invasion and migration via the AKT/CREB1 pathway [99]. Elevated GDF15 levels also reduce anti-tumor immunity by modulating NF-κB signaling and tumor-associated macrophage (TAM) activity [100]. Serum GDF15 demonstrated diagnostic sensitivity and specificity comparable to carbohydrate antigen 19-9 (CA 19-9) in distinguishing malignancies from benign neoplasms and healthy individuals [101] and even surpasses CA 19-9 in differentiating PC from chronic pancreatitis [102]. Moreover, elevated GDF15 serum levels are a sensitive biomarker for detecting PC in asymptomatic, high-risk individuals [103] and predicting survival outcomes [104].

In hepatocellular carcinoma (HCC), GDF15 promotes angiogenesis via Src signaling [105] and suppresses anti-tumor immunity by enhancing immune cell suppressive functions [106]. It has been recognized as a valid diagnostic and prognostic biomarker [106]. Serum GDF15 levels are significantly higher in patients with HCC and cirrhosis than in healthy controls, and their combination with α-fetoprotein (AFP) improves diagnostic accuracy [107]. Chen et al. reported elevated GDF15 levels in advanced HCC stages (Barcelona clinic liver cancer [BCLC] B and C) compared to early-stage disease (BCLC-A) [108]. GDF15 is also a predictor of HCC development in chronic HCV-related hepatitis patients who achieved a sustained virological response with direct-acting antivirals (DAAs) [83]. Despite evidence linking GDF15 to metabolic-associated fatty liver disease (MAFLD) and liver fibrosis [109], its role in HCC development is ambiguous, with studies suggesting both pro-tumorigenic and anti-tumorigenic effects [110,111].

**Table 1 biomedicines-13-01931-t001:** Overview of GDF15’s potential biological role across various tumor types.

Tumor Type	Tumorigenic Functions	Sample Type	Diagnostic Role	Prognostic Role	Predictive Role	References
CRC	Reduction in tumor lymphocyte infiltration and PD-L1 positivity. Promotion of invasion, EMT, and metastasis via TGF-β, Smad2/3, and PI3K/AKT/GSK-3β pathways. Enhancement of chemoresistance and oxidative stress control through Nrf2 activation.	Serum	Biomarker for CRC diagnosis. Biomarker for detection of adenomas and CRC recurrence. Association with metastasis.	Association with poor survival outcomes.	Association with resistance to oxaliplatin.	[79,86,87,88,89,90]
GC	Promotion of tumor growth and invasion via STAT3 phosphorylation and ErbB2 transactivation. Enhancement of chemoresistance through antioxidant pathways.	Serum	Biomarker for GC diagnosis.	Association with poor survival outcomes.	Association with resistance to cisplatin.	[80,91,92,93,94,95]
		Tissue	-	Association with better OS, while GFRAL and RET are associated with poor OS.	-	
PC	Promotion of tumor metastasis and chemoresistance via GDF15-GFRAL, NR5A2, p38 MAPK, and AKT/CREB1 pathways. Inhibition of immune response via NF-κB pathway and TAM activity.	Serum	Biomarker for PC diagnosis. Biomarker to differentiate PC from benign disease, chronic pancreatitis, and healthy individuals.	Association with PC progression, recurrence, and poor survival outcomes.	-	[96,97,98,99,100,101,102,103,104]
		Tissue	*-*	Association with PC progression.	-	
BC	Promotion of tumor EMT and metastasis through downregulation of E-cadherin and upregulation of mesenchymal markers. Enhancement of chemoresistance via HER2 and p38 MAPK phosphorylation. Maintenance of cancer stem cells through the ERK1/2 pathway. Induction of hepcidin via the SMAD1-5-8 pathway. Promotion of tumor sphere formation.	Tissue	-	-	Association with resistance to trastuzumab, eribulin, and paclitaxel (in TNBC). Association with radioresistance.	[112,113,114,115,116,117,118,119]
OC	Promotion of tumor proliferation, invasion, and chemoresistance. Downstream factor of GPX3.	Serum	Biomarker for OC diagnosis. Biomarker for detection of OC recurrence	Association with poor survival outcomes.	Association with first-line cisplatin resistance.	[81,120,121,122,123]
		Tissue	-	Association with poor survival outcomes.	GPX3 (promoting expression of GDF15) associated with ICI resistance. Correlation with PD-L1 expression.	
CCa	Promotion of tumor proliferation through the upregulation of cyclinD1/E1 and downregulation of p21 via the PI3K/AKT and MAPK/ERK pathways in complex with ErbB2.	*-*	-	-	-	[124]
PCa	Promotion of tumor metastasis via the FAK-RhoA pathway. Enhancement of chemoresistance. Promotion of bone metastasis through osteoblastic production of CCL2 and RANKL.	Serum	Biomarker for distinguishing PCa from benign conditions. GDPP is a diagnostic biomarker for bone metastasis in castration-resistant PCa.	Biomarker for distinguishing indolent and aggressive PCa. Predictor of poor cancer-specific survival.	Association with resistance to docetaxel in metastatic castration-resistant PCa	[125,126,127,128,129,130]
HCC	Modulator in MASH, MAFLD, liver fibrosis, and HCC. Conflicting roles in HCC development, with evidence suggesting both pro-tumorigenic and anti-tumorigenic effects. Promotion of angiogenesis via Src. Suppression of anti-tumor immunity.	Serum	Biomarker for HCC diagnosis. Biomarker for HCC recurrence post-DAA therapy in HCV+ patients.	Association with poor survival outcomes. Correlation with the severity of MAFLD and fibrosis progression.	-	[105,106,107,108,109,110,111]
NSCLC	Regulation of proliferation, invasion, and migration via PTEN/PI3K/AKT signaling pathway.	Serum	Biomarker for lung cancer diagnosis.	Associated with poor survival outcomes in locally advanced NSCLC treated with chemoradiotherapy.	Association with resistance to chemotherapy and ICIs.	[84,131,132]
Melanoma	GDF15 overexpression is a product of the constitutively active mutant V600E B-RaF. Induction of angiogenesis, role in melanocyte differentiation and proliferation.	Serum	-	Association with poor survival outcomes.	Association with ICI resistance.	[133,134]
OSA	Promotion of tumor metastasis via TGF-β signaling.	Serum and tissue	-	Association with poor survival outcomes.	-	[85]
GBM	Correlation with a reduction in tumor lymphocyte infiltration.	Tissue	-	Association with poor survival outcomes.	-	[135]
OSCC	Promotion of EMT via SMAD2/3, PI3K/AKT, and MEK/ERK pathways. Anti-apoptotic effects through interaction with p53.	Serum	Biomarker for distinguishing OSCC, oral leukoplakia, and healthy controls.	Association with poor prognosis. Association with persistence of disease after surgery.	Association with resistance to TPF induction treatment. Association with radioresistance.	[136,137,138,139]

In breast cancer (BC), GDF15 expression correlates with high tumor grade, estrogen receptor (ER) negativity, and human epidermal growth factor receptor 2 (HER2) positivity. It facilitates EMT and tumor invasion by downregulating E-cadherin and upregulating mesenchymal markers [112]. GDF15 also plays a role in iron metabolism dysregulation, cancer stem cell maintenance, and tumor spheroid formation [113,114]. In terms of therapeutic implications, GDF15 mediates HER2 and p38 MAPK phosphorylation, reducing trastuzumab sensitivity [115,116]. It is also overexpressed in paclitaxel-resistant triple-negative BC, correlating with poor prognosis [117], and promoting radioresistance through enhanced EMT and cancer stem cell-like properties [118]. Interestingly, Gkretsi et al. reported that GDF15 may suppress BC cell invasion via modulation of focal adhesion gene expression, suggesting a context-dependent role [119].

In ovarian cancer (OC), GDF15 levels are significantly elevated compared to those found in healthy controls and in patients with benign or borderline tumors [120]. Increased GDF15 expression is associated with advanced disease, poor survival, and cisplatin resistance [121,122]. GDF15 contributes to chemoresistance via modulation of canonical pathways in wild-type p53 cancer cells within the TME [123]. GDF15 expression is regulated by glutathione peroxidase 3 (GPX3), which supports pro-tumorigenic transcriptional programs and correlates with immune cell infiltration and PD-L1 expression [81]. In cervical cancer (CCa), GDF15 promotes tumor proliferation by upregulating CyclinD1 and CyclinE1 while downregulating p21 via the PI3K/AKT and MAPK/ERK pathways in conjunction with ErbB2 [124].

In prostate cancer (PCa), GDF15 outperforms prostate-specific antigen (PSA) in distinguishing PCa from benign prostate hyperplasia and healthy controls [125]. In patients with localized disease, elevated GDF15 levels are significantly associated with more aggressive behavior. High serum GDF15 levels are independently associated with poor cancer-specific survival, increased aggressiveness, and metastatic risk [126]. Overexpression of GDF15 promotes invasion and metastasis by activating actin cytoskeleton rearrangement through the focal adhesion kinase (FAK)-RhoA pathway [126]. GDF15 also confers cytoprotective effects, enabling PCa cells to survive in a hostile microenvironment, and its upregulation correlates with chemoresistance [82]. In metastatic castration-resistant PCa, elevated circulating GDF15 levels predict poor response to docetaxel treatment [127]. Interestingly, conflicting findings by Husaini et al. suggest that GDF15 may have immunostimulatory and anti-tumorigenic effects under certain conditions [128]. In the context of PCa bone metastases, osteocyte-derived GDF15 promotes tumor growth by upregulating early growth response 1 (EGR1) expression through the GFRAL pathway [65]. In addition, GDF15 propeptide (GDPP), secreted by osteoblasts, osteoclasts, and PCa cells, contributes to the TME of bone metastases by promoting proliferation, invasion, and migration. GDPP also serves as a novel biomarker for bone metastases in castration-resistant PCa [129,130].

In lung cancer, GDF15 is a potential diagnostic biomarker, with significantly higher serum levels compared to those with pneumonia or healthy controls. GDF15 shows superior diagnostic accuracy compared to conventional markers such as CEA, neuron-specific enolase (NSE), and cytokeratin fragment 21-1 (CYFRA 21-1). Elevated serum GDF15 levels are associated with advanced disease progression, poor clinical outcomes, and reduced chemotherapy efficacy [131]. In locally advanced non-small cell lung cancer (NSCLC) treated with chemoradiotherapy, elevated GDF15 levels correlate with worse recurrence-free survival (RFS) and OS, highlighting its prognostic relevance [84]. In metastatic NSCLC, elevated GDF15 expression is associated with immune evasion through modulation of the TME, characterized by reduced CD8+ T-cell infiltration, increased Treg cells, and resistance to ICIs [132].

In melanoma, it has been demonstrated that tissue GDF15 overexpression is probably a product of the ^V600E^B-Raf mutation in vitro and it influences melanocyte differentiation, proliferation, and angiogenetic processes. As a result, GDF15 has been shown to have undetectable tissue levels in normal melanocytes, whereas more than half of melanoma cells express it at high concentrations [133]. High GDF15 serum levels were also associated with poor prognosis [134], and it has been hypothesized that it might impair durable tumor control in patients treated with anti-CTLA-4 and be predictive of failure to ICIs [134]. In osteosarcoma (OSA), GDF15 promotes metastasis by regulating the TGF-β signaling pathway. Serum GDF15 levels have been suggested as a potential prognostic and predictive biomarker for lung metastasis [85].

In glioblastoma (GBM), high GDF15 expression correlates with reduced tumor-infiltrating lymphocytes. Downregulation of GDF15 increases T-cell infiltration and is related to prolonged survival [135].

Among the cancers arising from the head and neck area, there is some data about the role of GDF15 exclusively in a specific subsite, such as the oral cavity. Indeed, the GDF15 role in oral squamous cell carcinoma (OSCC) is discussed below.

Although GDF-15 has been investigated as a potential diagnostic biomarker in several malignancies [87,88,91,119,125], its clinical utility remains limited by substantial variability and lack of standardized thresholds. Elevated circulating GDF-15 is not specific for cancer or CC, as levels can be influenced by inflammation, cardiovascular and renal dysfunction, aging, and exposure to various medications. Therefore, while GDF-15 represents a promising marker of disease burden and systemic stress, current evidence is insufficient to support its use as a standalone diagnostic tool, and further prospective studies with rigorous validation across diverse populations are warranted.

## 4. A Focus on the Diagnostic, Prognostic, and Predictive Role of GDF15 in HEAD and Neck Cancer

As for other malignancies, GDF15 demonstrated pro-tumorigenic activity in HNC, by enhancing migration, invasion, and proliferation of HNC cells through different molecular mechanisms.

GDF15 is a TP53 transcriptional target and could be activated by the wild-type p53 protein thanks to the presence of two specific binding sites localized on the GDF15 gene promoter. Although their relation remains unclear, this may explain the anti-apoptotic effect of GDF15 overexpression in OSCC cell lines [136].

A recently discovered key process in HNC cancerogenesis is the positive feedback loop between GDF15 and the early growth response 1 (EGR1) gene; GDF15 promotes tumor progression by directly activating EGR1 and performing a paracrine regulation of its expression through phosphorylation of ERK1/2 and AKT; conversely, EGR1 actively binds the GDF15 gene promoter, resulting in GDF15 overexpression. It has also been shown that GDF15 modulates EMT via canonical (SMAD2/3) and non-canonical (PI3K/AKT and MEK/ERK) signaling pathways [137].

In fact, Zhang et al. reported an association between high serum GDF15 levels and early detection of OSCC and poor survival [138]. Similarly, elevated GDF15 plasma levels were found in oral leukoplakia, a high-risk precancerous lesion of the oral epithelial mucosa, and in OSCC [139].

Furthermore, GDF15 has shown a correlation with pathological tumor grading, since higher tissue levels, both in vitro (OSCC cell lines) and in vivo (tissue samples from OSCC patients), were associated with higher tumor grading, supporting the hypothesis that GDF15 is involved in promoting dedifferentiation in OSCC via activation of ERK1/2 and p90RSK [140]. Further evidence confirmed that tissue GDF15 levels were related to grade, presence of lymphovascular invasion, and higher tumor stages [137].

To further explore the molecular mechanisms leading to the pathogenetic function of GDF15, a study analyzed the genetic profile of OSCC patients using Next Generation Sequencing (NGS). Mutations of the GDF15 gene have been reported in nearly half of OSCC patients (41.3%). Missense mutations, occurring in the propeptide region involved in the correct folding and dimerization of GDF15, appear to be involved in excessive storage of GDF15, which thus could crosstalk with other oncogenic signaling pathways (such as Akt, ErbB2, and mTOR) and have been identified as an independent negative prognostic factor in OSCC patients [136].

In vivo experiments attempted to define the clinical usefulness of measuring tissue and serum GDF15 levels. Shiegnitz et al. examined serum GDF15 from OSCC patients before treatment and healthy controls: as previously demonstrated in other malignancies (e.g., CRC and prostate cancer), serum GDF15 concentration was significantly higher in OSCC patients. In contrast with other studies, GDF15 levels were not influenced by clinical risk factors such as tobacco and alcohol habits, nor by tumor size, tumor stage, and cervical lymph node metastasis, but potentially correlated with the grade of differentiation [141]. Serum GDF15 dynamics before and after surgery have been shown to be predictive of survival and could be used as a biomarker; stratifying patients according to preoperative serum GDF15 concentration (cut-off 875 pg/mL), it has been demonstrated that patients with lower GDF15 values had a significantly greater 5-year survival rate [141].

Investigating the genetic profile in nasopharyngeal carcinoma (NPC), it was found that gp96 and GDF15 were markedly upregulated in radioresistant tumor cell subclones and that the knockdown of GDF15 resulted in the restoration of radiosensitivity in vitro [142]. Overexpression of GDF15 has been reported to be associated with radioresistance also in OSCC, preventing cell apoptosis induced by reactive oxygen species (ROS) levels via a SMAD1-associated pathway [143,144]. Preclinical evidence suggested that the activation of the GDF-15/GFRAL/RET pathway increases the β-adrenergic signaling-dependent production of tissue-protective hepatic triglycerides, promoting a protective metabolic adaptation to an inflammatory microenvironment, as induced by the treatments [145]. These data underscore the predictive role of GDF15 in terms of response to radiotherapy, but also the therapeutic use of GDF15 inhibitors as radiosensitizing agents.

Regarding systemic treatments, a randomized phase III clinical trial investigated the predictive role of GDF15 in patients with locally advanced OSCC (stage III-IVA) treated with TPF (docetaxel, cisplatin, fluorouracil) as induction therapy. Comparing the two arms of the study, induction treatment with TPF did not improve survival outcomes, even when patients were stratified according to serum GDF15 levels. A long-term analysis of the clinical trial, with a median follow-up of 67 months, confirmed that low GDF15 levels were related to higher survival rates. Furthermore, patients with higher baseline tumoral GDF15 levels and cN0 had a significantly pronounced survival benefit from the TPF induction chemotherapy [146,147].

## 5. Targeting GDF15 as a Therapeutic Strategy in Cancer

In recent years, through the acquisition of new knowledge about the importance of GDF15 in CC and other carcinogenic processes, we are witnessing the development of target drugs that directly inhibit this molecule or its receptor GFRAL.

In mouse models of cancer cachexia, the antibody-mediated inhibition of GDF15-GFRAL has been shown in distinct studies to reverse the loss of adipose and muscle mass of tumor-bearing subjects [148], to improve muscle function and physical performance [149], and to reduce chemotherapy-induced weight loss [150].

Ponsegromab is a highly selective, humanized monoclonal antibody that inhibits the interaction between circulating GDF15 and GFRAL receptors. In a small phase Ib study enrolling patients with CC and elevated serum GDF15 levels, ponsegromab treatment reduced circulating GDF15 and was associated with increased appetite, weight gain, and improved physical activity [151]. Based on these preliminary findings, a phase 2, placebo-controlled trial (the PROACC-1 study) was conducted to evaluate the efficacy of ponsegromab (100, 200, or 400 mg) in terms of mean change in body weight at week 12 from baseline in patients with NSCLC, pancreatic, and colorectal cancer with cachexia and baseline GDF15 serum concentration ≥ 1500 pg/mL. Treatment was administered over a 12-week period [152]. The study met its primary endpoint (mean change in body weight at 12 weeks) across all ponsegromab-treated groups, with the most pronounced effect observed in patients receiving 400 mg, who achieved over 5% weight gain. Weight gain correlated with a decline in serum GDF15 and improvements in appetite, overall physical activity, and skeletal muscle mass. Adverse event rates were similar between the ponsegromab and placebo groups [153]. However, these results should be interpreted cautiously, given the relatively short treatment duration, the small sample size, and the heterogeneity of the study population. Additionally, long-term safety, durability of benefit, and impact on survival remain to be established in larger trials. Despite these limitations, the GDF15 pathway inhibition has shown interesting results.

Currently, other molecules targeting the GDF15 signaling pathway have been developed, with varying results [150,154,155], and other clinical trials are ongoing (summarized in Table 2).

GDF15-targeted therapies may also have potential in combination with anticancer treatments, including chemotherapy, immunotherapy, and radiotherapy. For example, a retrospective study including metastatic NSCLC patients treated with the anti-PD-1 antibody nivolumab found that low tumor GDF15 expression was associated with OS, PFS, and ORR, suggesting that GDF15 may identify subsets of patients less likely to benefit from immunotherapy [156]. In a phase 2 trial, visugromab, a humanized monoclonal antibody neutralizing GDF15, combined with nivolumab, demonstrated increased and durable responses in several heavily pretreated solid tumor cohorts and appeared to improve weight gain in patients with manifest cachexia [157]. Nevertheless, these findings derive from early-phase studies with limited patient numbers and require further validation. Visugromab is also being investigated in combination with nivolumab versus nivolumab plus placebo in muscle-invasive bladder cancer in the neoadjuvant setting, with the primary aim of evaluating the proportion of pathological complete response rates (ypT0N0) and radiologic responses [157]. Preclinical models further suggest that GDF15 inhibition might enhance the activity of bispecific T-cell engagers such as tebentafusp, which binds gp100 on GDF-15-secreting HLA-A2+ SK-MEL-5 melanoma cells, by restoring T-cell infiltration and improving tumor-specific immune responses [158].

In addition, GDF15 neutralization has been proposed as a strategy to mitigate chemotherapy-induced side effects, particularly platinum-related emesis and anorexia [159].

The inhibition of GDF15 signaling, while promising for alleviating CC and enhancing anticancer immunity, also raises important safety considerations. GDF15 is a pleiotropic cytokine involved in stress response signaling, appetite regulation, and metabolic adaptation under conditions such as infection, tissue injury, or chemotherapy-induced damage [66,156]. Its neutralization may increase susceptibility to exaggerated inflammatory reactions or impaired recovery after tissue damage. Moreover, GDF15 physiologically suppresses appetite and promotes weight loss via GFRAL-mediated signaling in the hindbrain [61,62]; thus, its blockade can lead to hyperphagia, rapid weight gain, and potential metabolic dysregulation, including impaired glucose homeostasis or exacerbation of insulin resistance [108]. In addition, chronic suppression of GDF15 might interfere with mitochondrial adaptations and protective metabolic reprogramming in conditions of cellular stress [43,108]. These concerns highlight the need for careful evaluation of long-term safety and metabolic consequences in ongoing and future clinical trials.

Overall, while targeting GDF15 represents a promising investigational approach in cancer cachexia and potentially in combination with immunotherapy or chemotherapy, the available evidence remains preliminary. Further well-designed trials are needed to better define the therapeutic role, long-term safety, and optimal patient populations for anti-GDF15/GFRAL antibodies.

## 6. Discussion

The emerging role of GDF15 in oncology represents a paradigm shift in the management of both cancer progression and CC. The dual function of GDF15 as both a tumor-promoting factor and a key player in metabolic dysregulation makes it an attractive target for novel therapeutic strategies.

Despite decades of research, no biomarker or pharmacologic therapy for CC has been integrated into standard clinical practice. Several factors have contributed to this translational gap, including the complex and multifactorial pathophysiology of cachexia, which involves overlapping inflammatory, neuroendocrine, and metabolic pathways [11,14,18]. Therapeutic strategies targeting single mediators, such as myostatin or activin A inhibition, have shown encouraging effects on muscle mass in early-phase trials but ultimately failed to demonstrate consistent improvements in functional outcomes or survival in larger studies [21,23]. In addition, the lack of standardized diagnostic criteria and validated, clinically meaningful endpoints has complicated trial design and regulatory approval [1,3]. Variability in cachexia definitions, patient populations, and outcome measures has further limited the comparability and reproducibility of results across studies. In this context, targeting GDF15 has emerged as a promising strategy that may help overcome some of the limitations faced by previous cachexia interventions and address both metabolic dysfunction and treatment resistance.

One of the most compelling aspects of targeting GDF15 is its broad therapeutic potential. GDF15 inhibition has shown promise not only in mitigating cachexia-related morbidity but also in enhancing the efficacy of systemic treatments, including chemotherapy and immunotherapy. The PROACC-1 trial has already demonstrated that neutralizing GDF15 with ponsegromab leads to weight gain, increased appetite, and improved functional performance, underscoring the role of GDF15 as a central regulator of cancer-associated metabolic dysfunction [152]. Additionally, the combination of visugromab with nivolumab has yielded encouraging responses in patients with refractory solid tumors, suggesting that targeting GDF15 could potentiate the efficacy of ICIs [157].

From a clinical standpoint, managing cachexia is equally important both for improving quality of life [158] and enhancing therapeutic outcomes. Systemic inflammation is a cause of CC, metabolic alterations, and treatment intolerance, which collectively have a detrimental impact on OS. Through GDF15-targeted interventions, oncologists may be able to increase treatment tolerability and efficacy, thereby improving patient prognosis in both localized and metastatic settings. Additionally, GDF15 has demonstrated multiple applications in diagnostics, prognostics, and therapeutics. Although influenced by both oncological and non-oncological conditions, its use in combination with traditional biomarkers has proven valuable across various solid tumors. The measurement of circulating GDF15 levels plays a significant prognostic role and could guide treatment decisions in specific patient subgroups [84,91,119,125,133].

The role of GDF15 is of particular interest in HNC, as patients frequently present with cachexia or malnutrition at diagnosis, either due to the disease itself or treatment-related sequelae. These conditions are particularly detrimental in the curative setting, where maintaining full treatment dose intensity is essential for optimal outcomes. In this context, providing appropriate nutritional support is crucial—not only to address malnutrition and mitigate the progression of cachexia, but also to improve treatment tolerance and overall prognosis.

While these findings are encouraging, significant gaps remain in our understanding of the precise role of GDF15 across different tumor types and stages. Although high circulating levels of GDF15 are associated with poor prognosis, the mechanistic underpinnings of its involvement in tumor biology and cachexia require further elucidation. Another crucial aspect is the potential interplay between GDF15 and other metabolic regulators, including IL-6, TNF-α, and TGF-β family members. Understanding these interactions may unlock new therapeutic synergies and provide insight into the development of combination strategies that integrate GDF15 inhibition with existing treatment modalities.

Finally, long-term safety data on GDF15-targeted therapies are currently lacking [150,152,153]. Given the physiological role of GDF15 in metabolic adaptation and stress responses, prolonged inhibition may have unintended consequences on metabolic homeostasis and immune function. Future clinical trials should aim to assess not only efficacy, but also potential adverse effects related to systemic GDF15 suppression.

## 7. Conclusions

GDF15 has emerged as a multifaceted biomarker with significant implications in oncology, spanning diagnostics, prognostics, and therapeutics. Its role in both cancer progression and cachexia underscores its dual relevance as a tumor-promoting factor and a regulator of metabolic dysfunction. Targeting the GDF15-GFRAL axis has shown promising early results in mitigating cachexia symptoms and improving treatment responses, particularly in combination with chemotherapy and immunotherapy. However, despite its potential, critical knowledge gaps regarding its mechanistic pathways, biomarker stratification, and long-term safety remain. Future research should focus on refining patient selection criteria, optimizing therapeutic strategies, and assessing combination approaches to maximize clinical benefit. With continued investigation, GDF15 modulation may become a cornerstone of precision medicine, enhancing both survival and quality of life in patients with various types of cancer.

## Figures and Tables

**Figure 1 biomedicines-13-01931-f001:**
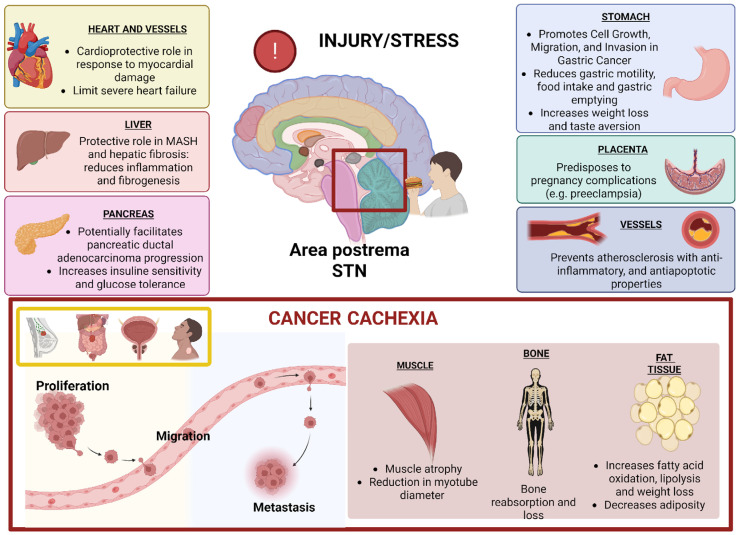
**The various implications in both pathological and physiological conditions of GDF15 [42,43,44,45,46,47]**. GDF15 has different physiological concentration gradients within the body, with the highest levels in placental tissue, the lowest in the kidney, liver, colon, pancreas, stomach, gallbladder, breast, lung, and endometrium, and intermediate levels in prostate and bladder. By binding GFRAL and exerting its action on the appetite areas located at the area postrema and nucleus of the solitary tract, GDF15 causes a significant reduction in food intake and weight and an increase in energetic expenditure. For these reasons, GDF15 is a major cause of CC syndrome characterized by anorexia, bone loss, adipose tissue depletion, muscle atrophy, and anemia. This is demonstrated by many preclinical studies on cellular lines and xenograft models and, for a second time, by the identification of higher levels of GDF15 in anorexic patients in comparison to ones with regular food intake. STN: Solitary tract nucleus.

**Figure 2 biomedicines-13-01931-f002:**
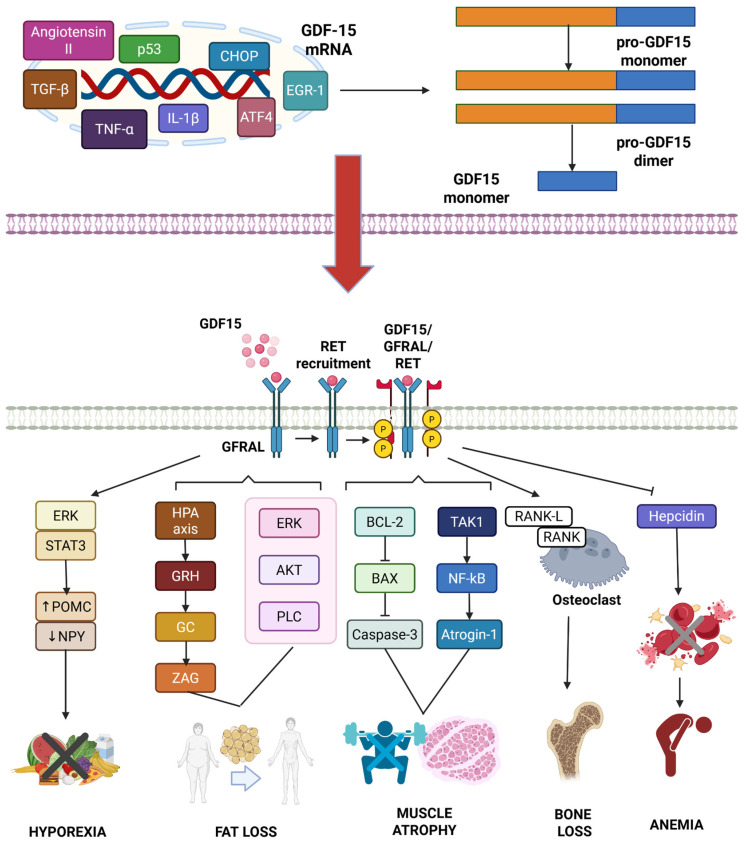
**Molecular pathways regulated by GDF15-GFRAL interaction [42,43,44,45,46,47]**. GDF15 expression is transcriptionally regulated by multiple stress-responsive and inflammatory mediators, including p53, EGR-1, CHOP, ATF4, interleukin-1β (IL-1β), TNF-α, angiotensin II, and TGF-β. Upon translation and maturation, GDF15 is secreted as a mature homodimer and binds to the GFRAL receptor in the area postrema and nucleus tractus solitarius of the brainstem. This interaction induces a conformational change in GFRAL that allows for the recruitment and dimerization of the co-receptor RET (rearranged during transfection). The activated GDF15–GFRAL–RET complex initiates downstream signaling cascades involving ERK, AKT, PLC, and STAT3. These signals contribute to anorexia through inhibition of NPY neurons and activation of POMC neurons, stimulate glucocorticoid release via HPA axis activation, and promote fat loss. GDF15 also induces muscle atrophy by activating pro-apoptotic and proteolytic pathways (e.g., BAX, Caspase-3, Atrogin-1) and enhances bone resorption through RANKL-mediated osteoclastogenesis. Moreover, GDF15-mediated hepcidin suppression is associated with anemia, likely due to impaired iron homeostasis and erythropoiesis.

**Table 2 biomedicines-13-01931-t002:** Ongoing/completed clinical trials involving GDF-15 and drugs targeting GDF-15 signaling pathway in patients with cancer.

Study Title	Phase	Drug	Study Population	Setting	Clinical Trial Number	Status	Results
GDF15-Based TPF Induction Chemotherapy for OSCC Patients	II	TPF (in patients with high GDF15 expression)	Advanced OSCC	Induction treatment	NCT02285530	Recruiting	NA
Study of the Efficacy and Safety of Ponsegromab in Patients with Cancer, Cachexia, and Elevated GDF-15 (PROACC-1)	II	Ponsegromab	NSCLC, pancreatic cancer, CRC	Cachexia and elevated serum GDF-15 concentrations	NCT05546476	Completed	Dose-dependent weight gain at 12 weeks (primary outcome), improved appetite and cachexia symptom control, increased physical activity, trends towards increased skeletal muscle mass (secondary endpoints)
First-in-Human Study of the GDF-15 Neutralizing Antibody Visugromab (CTL-002) in patients with Advanced Cancer (GDFATHER)	I/II	Visugromab + anti-PD-1	Advanced/metastatic solid tumors	Relapse/refractory to prior anti-PD-1/PD-L1 therapy	NCT04725474	Active, not recruiting	ORR 15% in NSCLC cohort, ORR 15% in bladder cancer cohort (PR/CR if PD-L1 TPS ≥ 5%), durable responses (>1 year), good safety
Neoadjuvant Immunotherapy Combined With the Anti-GDF-15 Antibody Visugromab to Treat Muscle-Invasive Bladder Cancer	II	Visugromab + Nivolumab	Urothelial carcinoma ineligible for cisplatin-based CHT	Neoadjuvant	NCT06059547	Recruiting	NA
A First-in-human Study to Evaluate the Safety and Tolerability of AZD8853 in Participants With Selected Advanced/Metastatic Solid Tumors	I/IIA	AZD8853	Advanced/metastatic NSCLC, MSS-CRC, urothelial carcinoma	Second line and beyond	NCT05397171	Completed (no further development planned)	No dose-limiting toxicity, 31.3% SD, 68.8% PD; no PR/CR. Transient GDF-15 suppression.
Study of NGM120 in Subjects With Advanced Solid Tumors, Pancreatic Cancer, and Prostate Cancer Using Combination Therapy	I/II	NGM120	Metastatic pancreatic adenocarcinoma, metastatic castration-resistant prostate cancer	First line and beyond	NCT04068896	Completed	36% of pts experienced >3.5% lean body mass gain by Week 9. In pancreatic cancer cohort: 2 pts with PR, 4 with SD; all 6 pts had ≥5% body weight gain and ~2.9% mean increase in lean body mass by Week 16
A Dose Escalation Study of AV-380 in Metastatic Cancer Patients with Cachexia	IB	AV-380	Metastatic CRC or pancreatic cancer	Patients with cachexia and elevated GDF-15 levels in the first-line setting with SoC	NCT05865535	Recruiting	NA

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
