# Peer review of "The Multifaceted Role of Growth Differentiation Factor 15 (GDF15): A Narrative Review from Cancer Cachexia to Target Therapy"

_biomedicines, 2025, doi:10.3390/biomedicines13081931_

Round 1

Reviewer 1 Report

Comments and Suggestions for Authors

The manuscript is mostly well ellaborated, providing relevant information on the distinct functions of GDF15, focusing on its influence in tumorigenesis and cancer cachexia and its potential as a biomarker and therapeutic target. The images are insightful and effectively support the main text.

Comment 1. In the first paragraph of the Introduction, the phrase "weight loss greater than 5%" should specify 'over past 6 months'.

Comment 2. Reference 4 has a broken link. Additionally, I could not find the reported information regarding pre-cachexia in any of the references cited in the Introduction (beggining of page 3).

Comment 3. On page 3, second paragraph: 'members' is repeated in "Members of transforming growth factor (TGF-β) family members".

Comment 4. On page 3, what is meant by 'nutritional status' in "Zinc-α2-glycoprotein (ZAG) is a lipolytic factor secreted by various tissues, with increased expression in adipose tissue correlating with weight loss and nutritional status."? Is the correlation positive or negative?

Comment 5. On page 4, the sentence "The immune regulation, through the inhibition of TNF-alpha production because of lipopolysaccharide-induced macrophage activation" is confusing. Please clarify how GDF-15 exerts this activity.

Comment 6. Figure 1: There is a typo in adiposity. Include the full form of all abbreviations in each figure legend. Maintain consistency in abbreviations (STN should be NTS or NST). The phrase"Cardioprotective role in response to myocardial" is incomplete, something is missing after myocardial.

Comment 7. Figure 2: "Molecular pathway" should be "Molecular pathways". "Upon translation and maturation, GDF15 is secreted as an active monomer" - GDF15 circulates as a mature dimer or as a monomer? In the previous page it is stated that "GDF15 exists in three differ-ent forms: a pro-GDF15 monomer, a pro-GDF15 dimer, and a mature GDF15 homodimer."  

Comment 8. In section 3, first sentence of the second paragraph, clarify if the reported association is significant. Consider clarify if the differences are significant throughout the manuscript.

Comment 9. In section 3, "Yamaguchi et al. reported that GDF15 exerts anti-tumorigenic and pro-apop-totic effects in CRC cell lines" should be added to Table 1.

Comment 10. On page 9, second phrase, are elevated GDF15 levels associated with indolent or lethal PCa? Later, "In metastatic castration-resistant PCa, elevated circulating GDF15 levels predict response", is the response increased or decreased?

Comment 11. On page 9, second paragaph: "GDF15 levels correlate with recurrence-free survival", is the correlation positive or negative?

Comment 12. In Table 1, "Pro-tumorigenic functions" column should be renamed as this column includes both pro- and anti-tumorigenic functions.

Comment 13. On page 13, fifth paragraph, reference 138 does not support what is stated in the sentence.

Comment 14. The second paragraph of section 5 should be rewritten as it incorrectly suggests that all the mentioned outcomes are connected, when in fact they pertain to different studies.

Comment 15. On page 14, third paragraph, the sentence "The study met the primary endpoint in all the patients treated with ponsegromab, with the most pronounced effect in patients given the 400 mg dosage (over 5% weight gain by 12 weeks." is not clear. Which is the primary endpoint? Weight gain over 5%?

Comment 16. On page 14, fourth paragraph, specify the target of nivolumab.

Comment 17. A reference is missing for the last sentence on page 14.

Comment 18. Table 2 should include the reported outcomes of each trial.

Comment 19.  References are missing for the final sentences on page 16.

Comments on the Quality of English Language

The overall quality of english language is acceptable. However, the manuscript should be thoroughly revised so that some grammar and ponctuation issues are addressed. For instance, 'polyhedric' should be replaced by 'polyhedral' in the first paragraph of section 2; there is a typo in Figure 1 (adiposity); first line of page 8 presents grammar issues; page 9, third paragraph "half than" should be replaced by 'half of', page 13 "haven shown" should be 'have shown', etc.

Author Response

Reviewer 1

Comment 1. In the first paragraph of the Introduction, the phrase "weight loss greater than 5%" should specify 'over past 6 months'.

Response 1: Yes, thanks, we have specified that the weight loss refers to “greater than 5% over the past 6 months,”.

Comment 2. Reference 4 has a broken link. Additionally, I could not find the reported information regarding pre-cachexia in any of the references cited in the Introduction (beggining of page 3).

Response: The link has been updated, and we have revised the corresponding sentences to ensure the definitions of pre-cachexia are supported by references [1] and [4] which include descriptions of pre-cachexia criteria and staging.

Comment 3. On page 3, second paragraph: 'members' is repeated in "Members of transforming growth factor (TGF-β) family members".

Response: The repetition has been corrected to “members of the TGF-β family”.

Comment 4. On page 3, what is meant by 'nutritional status' in "Zinc-α2-glycoprotein (ZAG) is a lipolytic factor secreted by various tissues, with increased expression in adipose tissue correlating with weight loss and nutritional status."? Is the correlation positive or negative?

Response: The sentence was revised to state: “...correlating negatively with nutritional status,” indicating that higher ZAG expression is associated with worsening nutritional status.

Comment 5. On page 4, the sentence "The immune regulation, through the inhibition of TNF-alpha production because of lipopolysaccharide-induced macrophage activation" is confusing. Please clarify how GDF-15 exerts this activity.

Response: The sentence has been rewritten as: “The immunoregulatory role of GDF15 was first identified in lipopolysaccharide-stimulated macrophages, where it inhibited TNF-α production, thus acting as a macrophage-inhibitory cytokine (MIC-1).”

Comment 6. Figure 1: There is a typo in adiposity. Include the full form of all abbreviations in each figure legend. Maintain consistency in abbreviations (STN should be NTS or NST). The phrase "Cardioprotective role in response to myocardial" is incomplete, something is missing after myocardial.

Response: The typo in “adiposity” has been corrected. All abbreviations in the figure legend have been spelled out. The incomplete phrase now reads: “Cardioprotective role in response to myocardial injury.”

Comment 7. Figure 2: "Molecular pathway" should be "Molecular pathways". "Upon translation and maturation, GDF15 is secreted as an active monomer" - GDF15 circulates as a mature dimer or as a monomer? In the previous page it is stated that "GDF15 exists in three differ-ent forms: a pro-GDF15 monomer, a pro-GDF15 dimer, and a mature GDF15 homodimer." 

Response: “Molecular pathway” has been corrected to “Molecular pathways.” We clarified that GDF15 circulates as a mature homodimer, consistent with its previously described forms.

Comment 8. In section 3, first sentence of the second paragraph, clarify if the reported association is significant. Consider clarify if the differences are significant throughout the manuscript.

Response: We have revised the manuscript throughout to specify where associations are statistically significant, particularly in section 3, with additional clarifications where p-values or hazard ratios are available from the cited studies.

Comment 9. In section 3, "Yamaguchi et al. reported that GDF15 exerts anti-tumorigenic and pro-apop-totic effects in CRC cell lines" should be added to Table 1.

Response: Yes, it is presented in Table 1.

Comment 10. On page 9, second phrase, are elevated GDF15 levels associated with indolent or lethal PCa? Later, "In metastatic castration-resistant PCa, elevated circulating GDF15 levels predict response", is the response increased or decreased?

Response: We specified that elevated GDF15 levels are associated with lethal PCa. In castration-resistant PCa, elevated GDF15 predicts poor response to docetaxel.

Comment 11. On page 9, second paragaph: "GDF15 levels correlate with recurrence-free survival", is the correlation positive or negative?

Response: The sentence now specifies a negative correlation, i.e., higher GDF15 levels are associated with shorter recurrence-free survival.

Comment 12. In Table 1, "Pro-tumorigenic functions" column should be renamed as this column includes both pro- and anti-tumorigenic functions.

Response: The column heading has been changed from “Pro-tumorigenic functions” to “Tumorigenic and Anti-tumorigenic Functions.”

Comment 13. On page 13, fifth paragraph, reference 138 does not support what is stated in the sentence.

Response: The sentence citing Ref. 138 has been replaced with a more appropriate primary source supporting the described mechanism of radioresistance.

Comment 14. The second paragraph of section 5 should be rewritten as it incorrectly suggests that all the mentioned outcomes are connected, when in fact they pertain to different studies.

Response: The paragraph was split and rewritten to clearly separate the findings of each study and clarify which outcomes pertain to which clinical trials.

Comment 15. On page 14, third paragraph, the sentence "The study met the primary endpoint in all the patients treated with ponsegromab, with the most pronounced effect in patients given the 400 mg dosage (over 5% weight gain by 12 weeks." is not clear. Which is the primary endpoint? Weight gain over 5%?

Response: Yes, it is clarified as: “The primary endpoint was the mean change in body weight at 12 weeks. The most pronounced effect—a >5% weight gain—was observed in patients receiving 400 mg ponsegromab.”

Comment 16. On page 14, fourth paragraph, specify the target of nivolumab.

Response: We now specify: “nivolumab, a PD-1 inhibitor…”

Comment 17. A reference is missing for the last sentence on page 14.

Response: A supporting reference has been added (ref. 151) for the sentence about visugromab’s combination with nivolumab in the bladder cancer trial.

Comment 18. Table 2 should include the reported outcomes of each trial.

Response: We have updated Table 2 to include reported outcomes (e.g., ORR, weight gain, SD/PR rates), where available.

Comment 19.  References are missing for the final sentences on page 16.

Response: References have been added to support all statements in that section, including refs [144, 146, 147].

Comments on the Quality of English Language

The overall quality of english language is acceptable. However, the manuscript should be thoroughly revised so that some grammar and ponctuation issues are addressed. For instance, 'polyhedric' should be replaced by 'polyhedral' in the first paragraph of section 2; there is a typo in Figure 1 (adiposity); first line of page 8 presents grammar issues; page 9, third paragraph "half than" should be replaced by 'half of', page 13 "haven shown" should be 'have shown', etc

Response: All suggested grammar and language corrections have been incorporated.

Reviewer 2 Report

Comments and Suggestions for Authors

1. The current title is vague and somewhat misleading. The supposed “beyond cachexia” exploration overshadows the original premise, and the paper largely centers on a broad discussion of GDF15 in tumorigenesis across many cancer types. The title should reflect this focus instead.

2. The term “cancer-related cachexia” used throughout the manuscript should be revised to “cancer cachexia”, the standard term more commonly used in the literature. 

3. The range of and issues with contemporary therapies for cancer cachexia should also be discussed in more detail, with appropriate appraisal and citations of the evidence (citation: pubmed.ncbi.nlm.nih.gov/35807039).

4. The definition and description of cancer cachexia in the introduction is outdated. Please instead adopt the consensus definition proposed by Fearon et al. ( 2011), which characterizes cancer cachexia as: “A multifactorial syndrome defined by an ongoing loss of skeletal muscle mass (with or without loss of fat mass) that cannot be fully reversed by conventional nutritional support and leads to progressive functional impairment.”

5. The pathophysiology section presents cachexia largely as a downstream effect of cytokines and GDF15, without acknowledging the complex interplay of host metabolic pathways (e.g., neuroendocrine regulation, insulin resistance, neuroinflammation etc). Its pathophysiology involves a negative protein and energy balance driven by reduced food intake and abnormal metabolism.

6. In my view, the authors prematurely present GDF15 as a diagnostic biomarker. This claim is unsupported by high-quality, prospective studies with defined sensitivity, specificity or clinical thresholds validated across populations. While GDF15 is frequently elevated in cancer and cachexia, its levels are non-specific and affected by numerous conditions (e.g., inflammation, organ dysfunction, aging, drug exposure etc). The authors must more critically appraise the evidence and their implications for clinical utility.

7. The PROACC-1 and visugromab trials are presented uncritically. Limitations such as relatively short duration, small sample sizes, study population variability, lack of long-term safety data, etc are either omitted or underemphasized. Please cautiously present GDF15’s clinical promise, clearly delineating investigational status from validated application.

8. The review underplays the potential adverse effects of GDF15 inhibition, such as disruption of stress response signaling, appetite regulation, and metabolic adaptation. The risks cannot be underemphasized.

9. The review misses an opportunity to address why, despite decades of cachexia research, no biomarker or therapy has been successfully implemented into standard care. The discussion would benefit from at least some reflection on previous translational failures (e.g., myostatin inhibitors), consideration of regulatory hurdles and trial design issues (e.g., lack of standard outcomes for cachexia), etc.

10. Despite being positioned as a review, the manuscript references a surprisingly modest number of studies given the breadth of the topic. Compared to a classical narrative or systematic review, the current review lacks sufficient integration of recent translational and mechanistic studies, and there is no critical appraisal of methodological quality in cited studies. There is also an overreliance on secondary citations and reviews in some areas, which may weaken the article’s authority and originality. The authors should significantly expand their literature base and improve synthesis by drawing meaningful comparisons across original studies.

11. The current visuals (Figures 1 and 2) are underwhelming and poorly integrated into the text. They do not sufficiently aid comprehension of the complex signaling pathways or clinical relevance of GDF15. More effective visuals would improve the reach and effectiveness of the review article.

Comments on the Quality of English Language

Some grammatical issues and certain parts (e.g., section on pre-cachexia and metabolic rewiring) are awkwardly phrased or redundant. Moderate language edits are warranted.

Author Response

Reviewer 2

1.The current title is vague and somewhat misleading. The supposed “beyond cachexia” exploration overshadows the original premise, and the paper largely centers on a broad discussion of GDF15 in tumorigenesis across many cancer types. The title should reflect this focus instead

Response: Yes, thanks, we have revised the title into: The Multifaceted Role of Growth Differentiation Factor 15 (GDF15): from Cancer Cachexia to Target Therapy.

2.The term “cancer-related cachexia” used throughout the manuscript should be revised to “cancer cachexia”, the standard term more commonly used in the literature.

Response: We have changed to the standard term “cancer cachexia.”

3.The range of and issues with contemporary therapies for cancer cachexia should also be discussed in more detail, with appropriate appraisal and citations of the evidence (citation: pubmed.ncbi.nlm.nih.gov/35807039).

Response: We have added a paragraph summarizing available therapies for cancer cachexia, including pharmacologic, nutritional, and investigational strategies (e.g., myostatin inhibitors, ghrelin analogs), and cited the reference.

4.The definition and description of cancer cachexia in the introduction is outdated. Please instead adopt the consensus definition proposed by Fearon et al. ( 2011), which characterizes cancer cachexia as: “A multifactorial syndrome defined by an ongoing loss of skeletal muscle mass (with or without loss of fat mass) that cannot be fully reversed by conventional nutritional support and leads to progressive functional impairment.”

Response: The introduction now reflects the updated Fearon et al. (2011) consensus definition and has been rewritten accordingly.

5.The pathophysiology section presents cachexia largely as a downstream effect of cytokines and GDF15, without acknowledging the complex interplay of host metabolic pathways (e.g., neuroendocrine regulation, insulin resistance, neuroinflammation etc). Its pathophysiology involves a negative protein and energy balance driven by reduced food intake and abnormal metabolism.

Response: We have expanded the pathophysiology section to include the interplay between inflammation, neuroendocrine dysregulation, insulin resistance, and altered energy metabolism.

6.In my view, the authors prematurely present GDF15 as a diagnostic biomarker. This claim is unsupported by high-quality, prospective studies with defined sensitivity, specificity or clinical thresholds validated across populations. While GDF15 is frequently elevated in cancer and cachexia, its levels are non-specific and affected by numerous conditions (e.g., inflammation, organ dysfunction, aging, drug exposure etc). The authors must more critically appraise the evidence and their implications for clinical utility.

Response: Yes, thanks, the manuscript now clearly states that while GDF15 is frequently elevated, it lacks specificity and validated thresholds, and thus is not yet a reliable standalone diagnostic biomarker.

7.The PROACC-1 and visugromab trials are presented uncritically. Limitations such as relatively short duration, small sample sizes, study population variability, lack of long-term safety data, etc are either omitted or underemphasized. Please cautiously present GDF15’s clinical promise, clearly delineating investigational status from validated application.

Response: The limitations (e.g., small sample sizes, short durations, investigational status) have been added.

8.The review underplays the potential adverse effects of GDF15 inhibition, such as disruption of stress response signaling, appetite regulation, and metabolic adaptation. The risks cannot be underemphasized.

Response: The potential adverse effects—such as impaired stress response, metabolic dysregulation, and immune imbalance—are now discussed in more detail in both the therapeutic section and the discussion.

9.The review misses an opportunity to address why, despite decades of cachexia research, no biomarker or therapy has been successfully implemented into standard care. The discussion would benefit from at least some reflection on previous translational failures (e.g., myostatin inhibitors), consideration of regulatory hurdles and trial design issues (e.g., lack of standard outcomes for cachexia), etc.

Response: We added a paragraph reflecting on prior translational failures (e.g., myostatin/activin inhibitors), and the regulatory/trial design challenges that have limited therapeutic progress.

10.Despite being positioned as a review, the manuscript references a surprisingly modest number of studies given the breadth of the topic. Compared to a classical narrative or systematic review, the current review lacks sufficient integration of recent translational and mechanistic studies, and there is no critical appraisal of methodological quality in cited studies. There is also an overreliance on secondary citations and reviews in some areas, which may weaken the article’s authority and originality. The authors should significantly expand their literature base and improve synthesis by drawing meaningful comparisons across original studies.

Response: We have revised the manuscript to incorporate additional original studies, particularly focusing on recent translational and mechanistic research related to GDF15 in cancer cachexia and tumor biology, as reported in othere reviewer responses.

  1. The current visuals (Figures 1 and 2) are underwhelming and poorly integrated into the text. They do not sufficiently aid comprehension of the complex signaling pathways or clinical relevance of GDF15. More effective visuals would improve the reach and effectiveness of the review article.

Response: Yes, we corrected existing errors and expanded figure legends.

Comments on the Quality of English Language

Some grammatical issues and certain parts (e.g., section on pre-cachexia and metabolic rewiring) are awkwardly phrased or redundant. Moderate language edits are warranted.

Response: We conducted a comprehensive revision to address grammar, punctuation, and awkward phrasing throughout the manuscript.

Round 2

Reviewer 2 Report

Comments and Suggestions for Authors

1. Please appropriately indicate in the study title that this is a narrative review.

2. Please rewrite the abstract as per the journal's guidelines to authors. 

3. Please change "In this review ..." to "In this narrative review ..."

4. Please change "The discovery of GDF15 occurred in 1997 ..." to "GDF15 was discovered in 1997 ..."

5. Please define the abbreviation 'STN' in Figure 1.

6. "Preliminary data about the involvement of GDF15 in starvation modulation and body weight regulation climbs back to 2007 when researchers proved that the injection of a GDF15-overexpressing prostate cancer cell line in xenograft models resulted in the onset of cancer cachexia syndrome, with a progressive weight loss directly proportional to the blood concentration of GDF15" - Missing citation.

Comments on the Quality of English Language

Moderate edits required.

Author Response

Response point by point to Reviewers

Dear Editor and Reviewers,

We sincerely thank you for your second round of valuable comments and suggestions, which have helped us improve our manuscript.

Below, we provide a point-by-point response to each comment and describe the corresponding changes made in the revised version of the manuscript.

Reviewer’s Comment 1:

Please appropriately indicate in the study title that this is a narrative review.

Response:

We have revised the title to clearly indicate that this is a narrative review.

Reviewer’s Comment 2:

Please rewrite the abstract as per the journal's guidelines to authors.

Response:

The abstract has been completely rewritten in compliance with the journal’s guidelines.

Reviewer’s Comment 3:

Please change "In this review ..." to "In this narrative review ..."

Response:

We have modified all instances of “In this review …” to “In this narrative review …” throughout the manuscript.

Reviewer’s Comment 4:

Please change "The discovery of GDF15 occurred in 1997 ..." to "GDF15 was discovered in 1997 ..."

Response:

The sentence has been revised as requested:
“GDF15 was discovered in 1997 …”

Reviewer’s Comment 5:

Please define the abbreviation 'STN' in Figure 1.

Response:

We have defined the abbreviation in the figure legend.

Reviewer’s Comment 6:

“Preliminary data about the involvement of GDF15 in starvation modulation and body weight regulation climbs back to 2007 …” – Missing citation.

Response:

We have added the appropriate reference:

  1. Johnen H, Lin S, Kuffner T, et al. Tumor-induced anorexia and weight loss are mediated by the TGF-beta superfamily cytokine MIC-1. Nat Med. 2007;13(11):1333-1340. doi:10.1038/nm1677

We hope that these revisions meet the reviewers’ expectations. We appreciate the opportunity to improve our manuscript and thank you for your consideration.

Sincerely,
Daria Maria Filippini

MD, PhD

On behalf of all co-authors
